# Plasma B Vitamers: Population Epidemiology and Parent-Child Concordance in Children and Adults

**DOI:** 10.3390/nu13030821

**Published:** 2021-03-02

**Authors:** Stephanie Andraos, Beatrix Jones, Clare Wall, Eric Thorstensen, Martin Kussmann, David Cameron-Smith, Katherine Lange, Susan Clifford, Richard Saffery, David Burgner, Melissa Wake, Justin O’Sullivan

**Affiliations:** 1The Liggins Institute, The University of Auckland, Auckland 1023, New Zealand; s.andraos@auckland.ac.nz (S.A.); e.thorstensen@auckland.ac.nz (E.T.); martin@kussmann.ch (M.K.); d.cameron-smith@auckland.ac.nz (D.C.-S.); 2Department of Statistics, Faculty of Science, The University of Auckland, Auckland 1010, New Zealand; beatrix.jones@auckland.ac.nz; 3Faculty of Medical and Health Sciences, The University of Auckland, Auckland 1023, New Zealand; c.wall@auckland.ac.nz; 4New Zealand National Science Challenge High-Value Nutrition, The University of Auckland, Auckland 1010, New Zealand; 5Singapore Institute for Clinical Sciences, Agency for Science, Technology and Research (A*STAR), 30 Medical Drive, Singapore 117609, Singapore; 6The Murdoch Children’s Research Institute, Parkville, VIC 3052, Australia; katherine.lange@mcri.edu.au (K.L.); susan.clifford@mcri.edu.au (S.C.); richard.saffery@mcri.edu.au (R.S.); david.burgner@mcri.edu.au (D.B.); melissa.wake@mcri.edu.au (M.W.); 7Department of Paediatrics, University of Melbourne, Parkville, VIC 3010, Australia; 8Department of Paediatrics, Monash University, Clayton, VIC 3800, Australia

**Keywords:** adults, B vitamins, children, growing up in Australia, longitudinal study of Australian children

## Abstract

Scope: B vitamers are co-enzymes involved in key physiological processes including energy production, one-carbon, and macronutrient metabolism. Studies profiling B vitamers simultaneously in parent–child dyads are scarce. Profiling B vitamers in parent–child dyads enables an insightful determination of gene–environment contributions to their circulating concentrations. We aimed to characterise: (a) parent–child dyad concordance, (b) generation (children versus adults), (c) age (within the adult subgroup (age range 28–71 years)) and (d) sex differences in plasma B vitamer concentrations in the CheckPoint study of Australian children. Methods and Results: 1166 children (11 ± 0.5 years, 51% female) and 1324 parents (44 ± 5.1 years, 87% female) took part in a biomedical assessment of a population-derived longitudinal cohort study: The Growing Up in Australia’s Child Health CheckPoint. B vitamer levels were quantified by UHPLC/MS-MS. B vitamer levels were weakly concordant between parent–child pairs (10–31% of variability explained). All B vitamer concentrations exhibited generation-specificity, except for flavin mononucleotide (FMN). The levels of thiamine, pantothenic acid, and 4-pyridoxic acid were higher in male children, and those of pantothenic acid were higher in male adults compared to their female counterparts. Conclusion: Family, age, and sex contribute to variations in the concentrations of plasma B vitamers in Australian children and adults.

## 1. Introduction

B vitamers play key roles in physiological processes such as macronutrient metabolism, nervous system functioning, hormone production, and one-carbon metabolism [1,2,3]. These metabolic pathways are central to the maintenance of metabolic health, and their dysregulation has been proposed to mediate the progression of adverse metabolic outcomes [4]. Circulating nutritional metabolites (e.g., B vitamers) are generally maintained within homeostatic ranges in healthy individuals [5]. However, food intake, disease status, physical activity, stress, body composition, and hormonal fluctuations can affect the levels of these metabolites [6,7,8,9,10,11]. Despite this, data on familial concordance, age- and sex-specificity of plasma B vitamers are limited [5,12]. Characterising homeostatic levels of B vitamers in large observational cohorts is paramount for outlining the epidemiological distribution of these micronutrients in healthy populations. 

B vitamers have generally been quantified separately in search of a specific deficiency in both clinical and research settings [5,6,8]. However, B vitamer deficiencies are generally uncommon in developed countries such as Australia, except for specific subgroups of the population (e.g., vegans, people affected by eating disorders, cancers, alcoholism, Crohn’s disease, and gastrointestinal infections) [13]. Over time, analytical advances have allowed for a simultaneous quantitation of inter-related nutritional metabolites in different body fluids [14,15,16,17], although this remains analytically challenging [18,19,20,21]. 

Given the fundamental roles that B vitamers play in health and disease, characterizing the epidemiology of these vitamers and their familial concordance is important as it may better inform us which factors contribute to variations in their circulating concentrations [1,2,3,4]. In this study, we applied ultra-high-performance liquid chromatography coupled with tandem mass spectrometry (UHPLC/MS-MS) to profile a selection of B vitamers simultaneously in the Growing Up in Australia’s Child Health CheckPoint. The Checkpoint study was a biomedical assessment nested between 2 waves (5 and 6) of the Longitudinal Study of Australian Children (LSAC), a national cohort that recruited individuals across all Australian states (i.e., New South Wales, Victoria, Queensland, South Australia, West Australia, Tasmania, Northern Territory, and Australian Capital Territory). We characterised (a) parent–child concordance, (b) generation-(parent versus child), (c) age-(within the adult subgroup of 28–71 years), and (d) sex-differences in circulating B vitamer concentrations. Considering that B vitamer concentrations are modulated by both endogenous (e.g., genetic, transcriptomic, metabolic) and environmental (e.g., diet, lifestyle) factors [1,2,3,4], we hypothesised that family (parent–child pairs) would explain a large proportion of variability of B vitamer levels.

## 2. Materials and Methods

### 2.1. Materials

Suppliers of materials were as detailed in Appendix A. All labelled and unlabelled standards listed in Appendix A were purchased from Sigma Aldrich. 

### 2.2. Ethical Approval, Consent and Sample Collection

The Royal Children’s Hospital (Melbourne, Australia) Human Research Ethics Committee (33225D), and the Australian Institute of Family Studies Ethics Committee (14–26) approved the Child Health CheckPoint study. During the 6th wave of the Longitudinal Study of Australian Children (LSAC), 3513 families agreed to be contacted for Checkpoint and 1874 parent–child dyads subsequently participated (53% uptake) [22,23] (Appendix A). Parents and caregivers provided consent for themselves and their children to participate in the CheckPoint study and for the collection and use of biological samples for research purposes.

### 2.3. Procedures

Venous blood was collected from parents and children into EDTA tubes, by a single venepuncture split to components including 6 plasma aliquots, following an average 4 h fast [24]. Blood samples were processed within ~1 h (1 min to 3.8 h). For the current analysis, 2490 EDTA plasma samples (1166 children and 1324 parents) were shipped, on dry ice in thermally monitored boxes, to the Liggins Institute, New Zealand. Sample tubes were randomised on dry ice onto 34 different 96-well FluidX plates, keeping parent–child pairs (1123 pairs) on the same plate, then stored at −80 °C prior to each assay.

### 2.4. Sample Preparation and Quality Control

Cold isotopically-labelled standards were weighed and diluted in either 0.1% HCl, 1:1 EtOH/H_2_O, 0.12% acetic acid/H_2_O, 5% of 0.1 mol/L NaOH, 20% EtOH in H_2_O, or H_2_O (Appendix A), these solutions were utilized to prepare the internal standard working solution (Appendix A), which was aliquoted (550 µL) into Eppendorf tubes, and stored (−80 °C, up to 4 months). One internal standard working solution aliquot was removed from the freezer at the start of the week, mixed with 4.45 mL deionized Millipore H_2_O (MilliQ^®^) and stored (+4 °C) for the remainder of the week. Unlabelled standards were diluted (Appendix A), aliquoted and stored individually (−80 °C). Unlabelled standards were not pre-mixed due to stability issues. Unlabelled standards were mixed to create a stock solution immediately prior to each assay. The concentrated stock (S10) was serially diluted (deionized H_2_O MilliQ^®^) to establish a gradient of concentrations (Appendix A). Subsequently, each standard solution (20 µL) was diluted in 180 µL of bovine serum albumin (4% BSA (*w*/*v*) in 0.01 M phosphate-buffered saline PBS) to mimic plasma.

To test for contamination and confirm the stability of the LC/MS-MS, a BSA solution (4% BSA (*w/v*) in 0.01 M phosphate-buffered saline (PBS)) was used in every assay. The system suitability test (SST) was performed using a mixture of standard S5 (20 µL), internal standard (10 µL), and 180 µL of deionized H_2_O (MilliQ^®^). The test standard was placed in the LC/MS-MS auto-sampler and the SST monitored to evaluate the consistency of peak ratios between labelled and unlabelled standards, peak shapes and chromatographic stability. Plasma samples from two volunteers were mixed to prepare a pooled (1:1) plasma quality control (QC). Three QCs were run in triplicates: A baseline QC (360 µL of pooled plasma), a second spiked QC (360 µL of plasma spiked with 40 µL S5), and a third spiked QC (360 µL of plasma spiked with 40 µL S6).

### 2.5. Sample Preparation and Robotic Automation

Randomized plates were thawed, in a room temperature water bath, before each assay run. Once completely thawed, the 96-well plate was mixed (gentle inversion) and centrifuged (500 g, 2 min, room temp). Protein precipitation was automated on an epMotion 5075vt robot (Eppendorf, Hamburg, Germany). Briefly, the robot was loaded with: (a) BSA (4% in PBS, 400 µL) as a blank; (b) standard solutions S1-S10; (c) all three plasma QCs; (d) 1 ml of the internal standard solution; (e) protein precipitation mixture (80 ml; 0.3% (*v/v*) acetic acid and 2.5% (*v/v*) MeOH); and (f) H_2_O (10 mL, deionized H_2_O (MilliQ^®^)). First, the protein precipitation mix (300 µL) was pipetted into all wells of the 96-well strata solid phase extraction protein precipitation plate (Phenomenex^®^). Second, the internal standard mix (10 µL) was dispensed into each well, excluding the first (blank). Then, a 100 µL aliquot of either QC, plasma sample, or standard was transferred into the appropriate well of the protein precipitation plate. The plate was placed on the robot’s thermomixer (room temperature; 800 rpm; 5 min), to facilitate protein precipitation. A vacuum was then applied to filter the samples into a new 96-well collection plate. The collection plate was dried (40 °C, 3 h, 0.03 bars of pressure) in a SpeedVac concentrator (Savant SC250EXP, Thermo Scientific, Waltham, MA, USA) coupled to a refrigerated vapour trap (Savant RVT4104, Thermo scientific, USA). Dried samples were then resuspended in a reconstitution solution (200 µL; ascorbic acid (1% *w/v*)) mixed with the mobile phase (5% acetic acid, (0.2% *w/v*)) of the ion pairing reagent heptafluorobutyric acid ((HFBA) in deionized H_2_O (MilliQ^®^)). The plate was mixed on the robot’s thermomixer (room temperature; 800 rpm; 5 min), then placed in the UHPLC autosampler and maintained at 10 °C throughout the assay run. 

### 2.6. UHPLC/MS-MS Assay

The analysis was performed on a Vanquish ultra-high-pressure liquid chromatography (UHPLC^+^) system, coupled with a TSQ Quantiva triple quadrupole mass spectrometer (Thermo Scientific). A Kinetex^®^ 2.6 µm F5 100 Å 150x2.1 mm column (Phenomenex^®^, Torrance, CA, USA) was used for chromatography. The mobile phase consisted of 5% acetic acid, 0.2% of the ion pairing reagent heptafluorobutyric acid (HFBA) in deionized H_2_O MilliQ^®^. A flow rate of 200 µL/min, starting at 4% acetonitrile and 96% mobile phase was applied. The run time for each sample was 14 minutes, with each full plate taking approximately 24 h to complete. Data processing was performed using Xcalibur version 4.0.27.19 (Thermo Scientific, USA). The calibration curves, quality controls, and peak integration were all assessed. The linearity of each standard calibration curve was checked, and the curves were used for quantification. Quality controls (QC) were used to monitor inter- and intra-assay reproducibility (with ≤20% variation between days being accepted). 

### 2.7. Statistical Analysis

All statistical analyses were performed in R programming environment version 3.6.1 [25]. Adjustments for plate effects were made for all compounds prior to normality checking to ensure that technical effects did not affect the distribution of vitamers. After adjusting for plate effects, histograms of all vitamers were plotted to check for normality. All compounds had skewed distributions (Figure 1) and therefore were log-transformed, and plate effects were removed on the log-scale using the *RANEF* function from the *lme4* package [26]. 

Two sets of mixed models were used to identify generation effects (parents versus children) and within family concordance using the *lme4* package on R [26], after adjusting for plate effects. Likelihood ratio tests were conducted between mixed models with and without the factor studied (i.e., generation or family). Effect sizes are summarized as the proportion of variation explained by the family effect, calculated as the ratio of family variance divided by the total variance of each adjusted/log-transformed variable. Pearson’s correlations were also performed within parent–child dyads to confirm familial correlations.

Two sets of linear models were used for each plate-adjusted/log-transformed variable to identify the effect of (a) sex in children and adults, and (b) age within the adult subgroup (given the wide age distribution of 28–71 years) using the *lme4* package on R [26].

Pearson’s correlation coefficients were used to test whether the plasma concentrations of riboflavin and flavin mononucleotide (FMN) correlated with each other. 

## 3. Result

### 3.1. Sample Characteristics

The analysis sample included 1166 children (52% girls, mean age 11 ± 0.5 years), and 1324 adults (87% mothers, mean age 44 ± 5.1 years) (Table 1). 

### 3.2. Plasma B Vitamers Are Weakly Concordant between Parents and Children 

Both likelihood ratio tests (Table 2) and Pearson’s correlations highlighted the strongest family effect to be for nicotinamide (correlation coefficient [confidence interval adjusted for multiple testing]: (0.19 [0.13; 0.24]), followed by pantothenic acid (0.16 [0.10; 0.22]); thiamine (0.09 [0.04; 0.16]); FMN (0.07 [0.01; 0.13]), riboflavin (0.05 [−0.01; 0.11]), and finally 4-pyridoxic acid (0.06 [0.0001; 0.12]) (Figure 2). Family effects explained 31% of the variability for nicotinamide, 17% for pantothenic acid, 13% for thiamine, 16% for FMN, 11% for riboflavin, and 9% for 4-pyridoxic acid (Table 2).

### 3.3. Plasma B Vitamer Concentrations Are Age-Dependent

Differences in plasma concentrations were identified between children and adults for all compounds except for flavin mononucleotide (FMN) (Table 3). Thiamine and nicotinamide had higher concentrations in children (median (lower quartile; upper quartile): 4.14 (2.74; 6.13) nM; 430.85 (285.61; 730.60) nM respectively) when compared to adults (2.14 (1.05; 4.11) nM; 396.41 (268.50; 640.74) nM respectively; *p* < 0.0001). By contrast, riboflavin, pantothenic acid, and 4-pyridoxic acid had markedly higher concentrations in adults (14.44 (9.24; 25.03) nM; 179.94 (140.02; 238.19) nM; 15.58 (9.24; 25.59) nM respectively) than children (14.24 (9.85; 21.2) nM; 173.72 (144.76; 206.79) nM; 11.72 (7.75; 17.50) nM respectively; *p* ≤ 0.001) (Table 3). Given the wide age distribution in the parent subgroup (28–71 y), we further investigated whether B vitamer concentrations differed with adult age. Riboflavin, pantothenic acid, and 4-pyridoxic acid (*p* ≤ 0.01 for all) exhibited positive associations with age in the adult subgroup (Figure 3 and Appendix A).

### 3.4. Plasma B Vitamer Concentrations Are Sex-Dependent 

Sex-differences in B vitamer concentrations were more evident in children compared to adults (Table 4). The concentrations of thiamine, pantothenic acid, and 4-pyridoxic acid were higher in male children (4.43 (2.88; 6.79) nM; 182.53 (153.72; 219.27) nM; 12.60 (8.18; 19.21) nM respectively) compared to female children (3.92 (2.69; 5.57) nM; 165.17 (138.68; 196.33) nM; 11.15 (7.36; 16.18) nM respectively; *p* ≤ 0.01) (Table 4). In adults, only pantothenic acid exhibited higher levels in males (196.89 (157.10; 250.52) nM) than females (175.96 (138.79; 233.43) nM; *p* = 0.02) (Table 4). By contrast, riboflavin, FMN, and nicotinamide were similar between the sexes in both generations. 

### 3.5. B2 Vitamers Are Positively Correlated

Plasma concentrations 0.0001), and children (R = 0.53, *p* < 0.0001) (Figure 4). 

## 4. Discussion 

B vitamer concentrations exhibited only weak familial concordance between children and parents in this Australian sample, therefore refuting our proposed hypothesis. Thiamine (B1) and nicotinamide (B3) plasma concentrations were higher in children than adults. In contrast, riboflavin (B2), pantothenic acid (B5), and 4-pyridoxic acid (B6) plasma concentrations were higher in adults than children. Plasma B vitamers exhibiting sex-specificity were higher in males than females in both children (for thiamine (B1), pantothenic acid (B5) and 4-pyridoxic acid (B6)) and adults (for pantothenic acid exclusively). Sex and age are important factors characterizing the levels of plasma B vitamers. Moreover, we contend that the weak parent–child concordance in B vitamer levels likely reflects a high inter/intra-individual variability in dietary and metabolic (e.g., excretory) determinants.

B vitamers are water-soluble micronutrients present in foods in either their free forms, bound to proteins, and/or as co-enzymes. B vitamers are generally highly bioavailable from foods (50–95% bioavailable within mixed diets), especially when in their co-enzymatic forms [13]. Their storage in the human body is minimal (except for vitamin B12), and their retention within tissues (i.e., the brain, liver, kidneys, spleen, and muscles) occurs in more complex forms (i.e., protein-bound or as co-enzymes). Excess water-soluble vitamins are readily excreted primarily in the urine, but also by the lungs, and faeces [13]. Therefore, B vitamers need to be constantly supplied by the diet to maintain the homeostatic levels required to activate the metabolic machinery [1]. Parent–child correlations in B vitamer concentrations were generally weak. Notably, the lowest variability explained by a shared gene–environment (family) effect was evident for B vitamers that reflect dietary intake in different body fluids (e.g., 9% for 4-pyridoxic acid, 11% for riboflavin, and 16% for FMN) [20,27,28,29]. Conversely, the B vitamer with the highest percentage of variability explained by family effects was nicotinamide (31%), which is not as representative of niacin intake [5,7,27,30]. Nicotinamide forms the bioactive coenzyme NAD (nicotinamide adenine dinucleotide) and its phosphorylated form NADP [5,7,27,30]. This pinpoints a weak dietary contribution to the shared gene–environment effect on plasma B vitamers in family settings, which may be due to a variability in dietary intakes between generations.

Thiamine (B1) and nicotinamide (B3) plasma concentrations were markedly higher in children compared to adults, whereas riboflavin (B2), pantothenic acid (B5) and 4-pyridoxic acid (B6) were higher in adults compared to children. It is possible that these differences reflect developmental requirements for these vitamers. However, proposing developmental differences in metabolic utilization as an explanation for these inter-generational variations may be misleading because: (1) B vitamers are involved in critical physiological metabolic networks across the lifecycle (e.g., the Krebs cycle, the electron transport chain, one-carbon metabolism), and (2) we have only characterised B vitamer profiles at one semi-fasted time point in these dyads, making impacts due to recent dietary intake a possibility [3,13]. Therefore, it remains likely that higher levels of B vitamers simply reflect differences in recent intake or excretion rates [20,28].

Differences between children and adults were evident for all compounds except for FMN. The riboflavin-FMN generational disparity is interesting as these metabolites belong to the same B2 family (Figure 5) [13]. In food, vitamin B2 is present as free riboflavin, FMN, and flavin adenine dinucleotide (FAD) [3]. Riboflavin is transported in the blood as free riboflavin, FMN bound to albumin, as well as a specific riboflavin-binding protein [3]. Cells take up free riboflavin from the circulation, convert riboflavin to FMN and then to FAD, and bind them to cellular proteins, inhibiting their diffusion out of the cells. Excess riboflavin (as unmodified, glycosylated, oxidised, de-methylated, or hydroxylated riboflavin) is excreted in the urine [3]. We observed higher plasma riboflavin levels in adults compared to children. FMN levels were not markedly different between generations but exhibited a trend towards higher levels in children. Since the vitamin B2 group is involved in the KREBS cycle and fatty acid redox reactions for energy production [3,13], higher FMN levels in children with lower baseline riboflavin levels may reflect an improved efficiency of metabolic utilisation of riboflavin, with increased demands during the growth phase. FMN may be less affected by generational riboflavin fluctuations, consistent with homeostatic maintenance of metabolically active B vitamers that are key players in vital pathways of energy production and antioxidant mechanisms [7,31,32]. These hypotheses are further supported by previous findings [29,31,32,33]. For example, plasma FMN was less responsive (27% plasma increase) to riboflavin supplementation than plasma riboflavin levels (83% increase) [29]. Moreover, following oral and intravenous riboflavin administrations, plasma flavo-coenzymatic variations were less pronounced than those of plasma riboflavin [33]. Validating the hypothesis of homeostatic maintenance of FMN and FAD in healthy settings would require the comparison of riboflavin, FMN, and FAD levels in intervention studies to identify differential regulatory pathways of homeostasis between healthy and unhealthy individuals.

Plasma concentrations of riboflavin, pantothenic acid, and 4-pyridoxic acid (4-PA) were positively associated with adult age (28–71 years). This is consistent with reports of age-specific changes in vitamin B6 status (i.e., decreases in pyridoxal 5’-phosphate (PLP) and/or plasma/urinary increases in its catabolite 4-PA) [20,34,35]. Age-dependent changes in B6 status have been proposed to be due to decreased intake, higher catabolic rates, increased enzymatic activity of alkaline phosphatase (responsible for PLP hydrolysis into pyridoxal (PL) and 4-PA), as well as variations in kidney function that impact B vitamer excretion [20,36,37]. The synthesis of PLP in tissues occurs via pyridoxine oxidation by FMN (riboflavin coenzyme) dependent enzymes (i.e., pyridoxine phosphate oxidase (PPO)) [37,38]. Since excess PLP is catabolized into PL then 4-PA, positive associations of 4-PA with age may reflect increased PLP turnover and catabolism with higher circulating riboflavin concentrations. Plasma pantothenic acid concentrations were also positively associated with age. Notably, plasma concentrations of pantothenic acid are more readily affected by recent dietary intakes compared to its bound forms found within erythrocytes (i.e., coenzyme A) [39]. Studies in rural Japan have reported a decline in erythrocyte pantothenic acid with age, and proposed a potential decreased capacity of synthesising bound pantothenic acid in older adults [39,40,41]. While results from exclusively rural areas may not be relevant to populations in Australia, it is plausible to suggest that recent dietary intake affects plasma pantothenic acid, or that higher plasma pantothenic acid may reflect a lower capacity of synthesising its bound form in the tissues of older individuals. These age-dependent differences highlight the need to account for age when profiling B vitamers and the potential for using metabolomics to characterise physiological changes underlying the ageing process. 

We identified sex-specific differences in some B vitamers in our study. One of the challenges in nutritional metabolomics is that it is unclear whether changes in blood metabolites are caused by the diet itself, by upregulation of dietary co-factors/co-enzymes involved in food metabolism, or by a combination of both [42,43,44,45,46]. Given the ubiquitous availability of these vitamers in foods [3], different B vitamer concentrations by sex and age may solely represent differences in the amounts of foods consumed [47,48], and/or an upregulation by feedback response of the enzymatic machinery involved in metabolic reactions associated with increased energy intake and production (e.g., the Krebs cycle, amino and fatty acid metabolism). Sexual dimorphism has been observed in several groups of nutritional metabolites (e.g., vitamin B12, vitamin C, vitamin D, vitamin B6, folates, lipids, amino acids, trimethylamine N-oxide (TMAO), betaine, choline, carnitine, dimethylglycine), and genetic variants have been associated with the status of some of these metabolites [46,49,50,51,52,53,54,55]. Differences in hormonal fluctuations, body composition, food intake, and growth rates may all explain sex-specific differences in circulating nutritional metabolites (e.g., B vitamers) [10,11,20,28,48,56]. Notably, sex-specific differences in B vitamers were generation-specific, with more evident differences in children compared to adults. This may simply be due to an unbalanced male to female ratio (1:10) in adults compared to children (1:1). Given the sex-specificity of plasma B vitamers, future epidemiological studies should aim to profile cohorts with a balanced male to female ratio. 

### Limitations

Given the difficulty in detecting and measuring some of the B vitamers in body fluids, several of our measures were below the limit of detection and were excluded from our analysis and discussion. Therefore, our final set of B vitamer measures do not fully reflect B vitamer status. Despite the large sample size (N = 2490 individuals), our study was cross-sectional and only a single semi-fasted plasma sample was collected from our participants. Moreover, males were underrepresented in the adult subgroup (1:10 ratio of males to females in adults compared to 1:1 male to female ratio in children), which limited our ability to identify sex-specific differences, and to examine the effect of age in adult males. Furthermore, our cohort is relatively wealthier than the general Australian population (over 78% of our population scored in the middle to least disadvantaged SEIFA score compared to around 62% in the general Australian population) [57,58]. Finally, accounting for additional drivers of B vitamer concentrations such as physical activity status and body composition could have strengthened some of our conclusions. However, these were not included as part of this analysis given that our aim was to identify the population distribution of circulating B vitamer concentrations in children and adults.

## 5. Conclusions

We have described nutritional profiles of the B vitamin complex in an Australian cohort of children and adults. We have provided evidence for differences in plasma B vitamer levels by sex and age, and also highlighted parent–child similarities, especially for nicotinamide. Parent–child concordance tended to differ by vitamer, which may reflect differential inter-conversions between metabolically active vitamers and those reflective of dietary intake. Collectively, our results highlight that sex and age are more important contributors to B vitamer concentrations than shared heritability in a family setting. Other underlying anthropometric and mechanistic factors that are age- and sex-specific such as body composition, hydration status, rates of urinary excretion, hormonal fluctuations, dietary intakes, and physical activity levels may begin to explain the epidemiological distributions of B vitamer concentrations and should be further investigated in future studies. The objective characterisation of B vitamer reference ranges in population settings of both children and adults will inform on vitamer homeostasis, and potentially improve the ability to identify their abnormal ranges. Future studies should evaluate associations between B vitamer profiles and metabolic outcomes in diseased versus healthy individuals to better characterise micronutrient profiles of normality

## 6. Patents

This section is not mandatory but may be added if there are patents resulting from the work reported in this manuscript.

## Figures and Tables

**Figure 1 nutrients-13-00821-f001:**
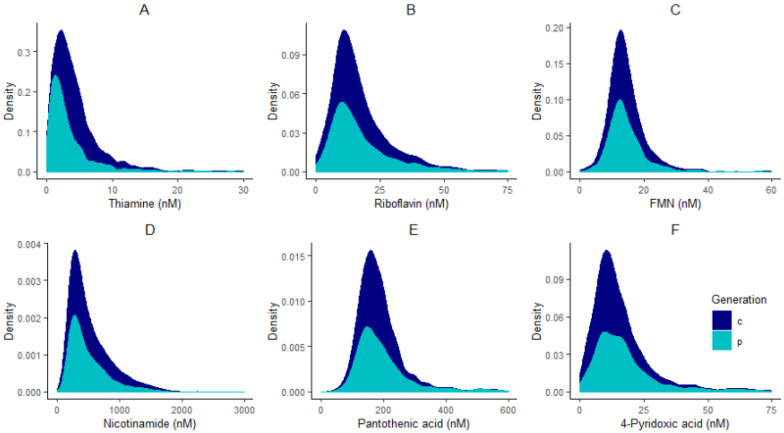
Distribution plots of thiamine (**A**), riboflavin (**B**), FMN (**C**), nicotinamide (**D**), pantothenic acid (**E**), and 4-pyridoxic acid (**F**) in children “c” and parents “p”.

**Figure 2 nutrients-13-00821-f002:**
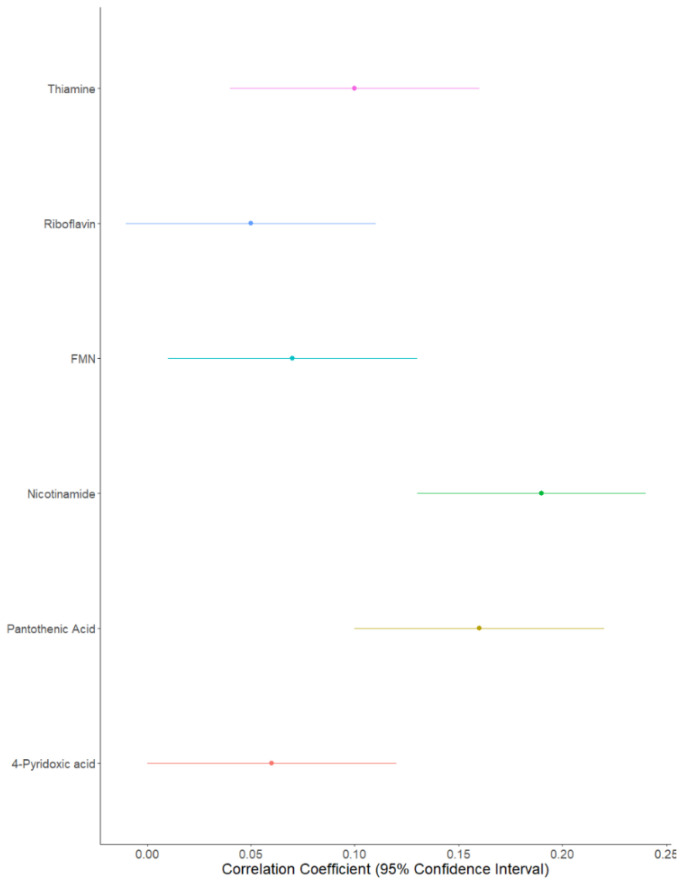
Forest plot of B vitamer correlations between parent–child pairs.

**Figure 3 nutrients-13-00821-f003:**
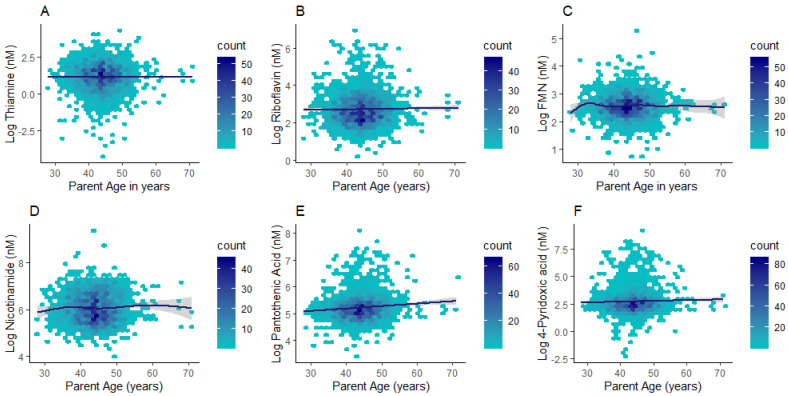
Hexagonal plots of thiamine (**A**), riboflavin (**B**), FMN (**C**), nicotinamide (**D**), pantothenic acid (**E**), and 4-pyridoxic acid (**F**) concentrations across the adult age range.

**Figure 4 nutrients-13-00821-f004:**
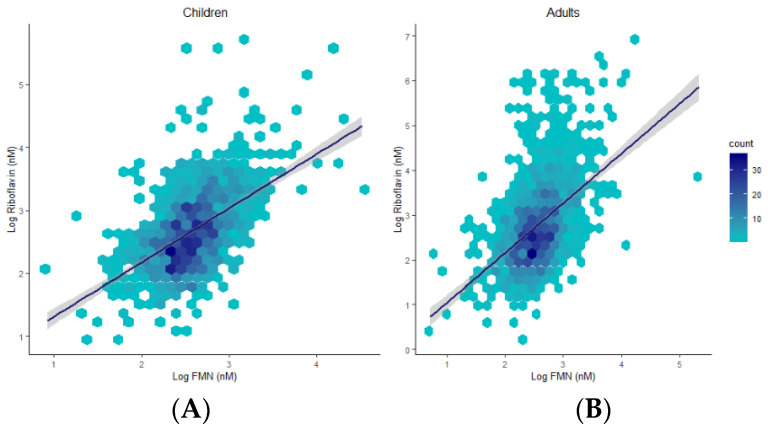
Hexagonal plots and trends of associations between riboflavin and FMN in children (**A**) and adults (**B**).

**Figure 5 nutrients-13-00821-f005:**
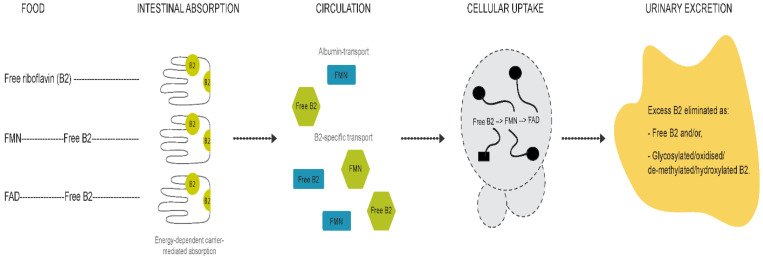
Graphical representation of vitamin B2 metabolism.

**Table 1 nutrients-13-00821-t001:** Sample characteristics.

	Children			Adults		
Characteristic	All	Male	Female	All	Male	Female
*N*	1166	565	601	1324	174	1150
Age in years (mean ± SD)	11.4 ± 0.5	11.4 ± 0.5	11.5 ± 0.5	43.9 ± 5.1	46.2 ± 6.4	43.6 ± 4.8
BMI rounded in kg/m^2^ Median (Lower-Upper Quartiles)	18.4 (16.8–20.6)	18.1 (16.7–20.2)	18.8 (17.0–21.1)	26.54 (23.4–31.0)	27.4 (25.2–31.1)	26.3 (23.1–31.0)
BMI Z-scores (mean ± SD)	0.3 ± 0.9	0.31 ± 0.94	0.31 ± 0.95	N/A	N/A	N/A
Biological parent of child (N)	N/A	N/A	N/A	1313	172	1141
Australian state of current residence: State (N)	New South Wales (359); Victoria (261); Queensland (221); South Australia (92); West Australia (139); Tasmania (40); Northern Territory (17); Australian Capital Territory (38)			New South Wales (391); Victoria (311); Queensland (240); South Australia (108); West Australia (164); Tasmania (46); Northern Territory (18); Australian Capital Territory (47)		
Socio-Economic Indexes for Areas (SEIFA) disadvantage Quintile (N)	Most Disadvantaged (83); Second Most (171); Middle (199); Second Least (272); Least Disadvantaged (442)			Most Disadvantaged (94); Second Most (193); Middle (233); Second Least (304); Least Disadvantaged (501)		

Skewed variables were reported as medians and lower/upper quartiles, and normally distributed variables as means and standard deviations.

**Table 2 nutrients-13-00821-t002:** Mixed model results, variances and effect sizes of family on B vitamer concentrations.

Vitamer	Effect of Dyad (or Family) on Mixed Model *	Family Effect Variance	Vitamer Variance	Effect Size of Family on Vitamer Concentrations (%)
Thiamine (B1)	*p* < 0.0001 Log likelihood with family = −2991.1 Log likelihood without family = −3004.2	0.12	0.82	13
Riboflavin (B2)	*p* < 0.001 Log likelihood with family = −2969.8 Log likelihood without family = −2976.7	0.07	0.64	12
FMN (B2)	*p* < 0.0001 Log likelihood with family = −1211.6 Log likelihood without family = −1224.9	0.02	0.16	16
Nicotinamide (B3)	*p* < 0.0001 Log likelihood with family = −2291.2 Log likelihood without family = −2230.9	0.11	0.37	31
Pantothenic acid (B5)	*p* < 0.0001 Log likelihood with family = −1488.3 Log likelihood without family = −1504.0	0.03	0.20	17
4-Pyridoxic acid (B6)	*p* = 0.003 Log likelihood with family = −3516.1 Log likelihood without family = −3520.5	0.10	1.11	9

* Log transformed variable for statistical modelling. Medians (LQ: Lower Quartile; UQ: Upper Quartile) reported for all variables.

**Table 3 nutrients-13-00821-t003:** Medians, Lower/Upper quartiles, and mixed model results of B vitamers in children and parents.

Vitamer (nM)	Adult Median (LQ, UQ) *	Child Median (LQ, UQ) *	Effect of Generation on Mixed Model (Children, Adults) **
Thiamine (B1)	2.14 (1.05; 4.11)	4.14 (2.74; 6.13)	*p* < 0.0001 Log likelihood with generation = −2991.1 Log likelihood without generation = −3164.4
Riboflavin (B2)	14.44 (9.24; 25.03)	14.24 (9.85; 21.2)	*p* = 0.001 Log likelihood with generation = −2969.8 Log likelihood without generation = −2975.3
FMN (B2)	13.27 (10.87; 16.86)	13.72 (11.18; 16.81)	*p* = 0.129 Log likelihood with generation = −1211.6 Log likelihood without generation = −1212.8
Nicotinamide (B3)	396.41 (268.50; 640.74)	430.85 (285.61; 730.60)	*p* < 0.0001 Log likelihood with generation = −2230.9 Log likelihood without generation = −2241.5
Pantothenic acid (B5)	179.94 (140.02; 238.19)	173.72 (144.76; 206.79)	*p* < 0.0001 Log likelihood with generation = −1488.3 Log likelihood without generation = −1513.0
4-Pyridoxic acid (B6)	15.58 (9.24; 25.59)	11.72 (7.75; 17.50)	*p* < 0.0001 Log likelihood with generation = −3516.1 Log likelihood without generation = 3595.0

* Log transformed variable used for statistical modelling. ** Medians (LQ: Lower Quartile; UQ: Upper Quartile) reported for all variables.

**Table 4 nutrients-13-00821-t004:** Medians, lower/upper quartiles and linear model results of B vitamer levels by sex in each generation.

Vitamer (nM)	Children				Adults			
	Females Median (LQ; UQ) *	Males Median (LQ; UQ)	Adjusted R^2^ of Linear Model ^+^	*p* Value	Females Median (LQ; UQ)	Males Median (LQ; UQ)	Adjusted R^2^ of Linear Model ^+^	*p* Value
Thiamine (B1)	3.92 (2.69; 5.57)	4.43 (2.88; 6.79)	0.01	<0.0001	2.09 (1.02; 4.06)	2.50 (1.17; 4.36)	0.001	0.19
Riboflavin (B2)	14.17 (9.80; 21.39)	14.27 (9.94; 21.20)	−0.001	0.70	14.56 (9.22; 25.01)	13.83 (9.50; 26.24)	−0.001	0.77
FMN (B2)	13.62 (10.99; 17.13)	13.87 (12.25; 16.61)	−0.00009	0.35	13.27 (10.83; 16.91)	13.22 (11.15; 16.20)	−0.001	0.79
Nicotinamide (B3)	432.21 (273.23; 730.65)	430.44 (297.09; 729.99)	−0.001	0.56	396.41 (264.47; 644.74)	398.41 (293.46; 613.64)	−0.0004	0.50
Pantothenic acid (B5)	165.17 (138.68; 196.33)	182.53 (153.72; 219.27)	0.025	<0.0001	175.96 (138.79; 233.43)	196.89 (157.10; 250.52)	0.003	0.02
4-Pyridoxic acid (B6)	11.15 (7.36; 16.18)	12.60 (8.18; 19.21)	0.006	0.01	15.58 (9.16; 25.48)	16.02 (10.31; 27.33)	−0.001	0.97

* LQ: Lower quartile; UQ: Upper quartile. Summary statistics are reported as medians and lower/upper quartiles from skewed variables. ^+^ Fitted with log-of FMN and riboflavin were strongly positively correlated in both adults (R = 0.49, *p* < transformed variables.

## Data Availability

Data described in the article will be made available upon request after application and approval by our teams.

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
