# Peer review of "Plasma B Vitamers: Population Epidemiology and Parent-Child Concordance in Children and Adults"

_nutrients, 2021, doi:10.3390/nu13030821_

Round 1
Reviewer 1 Report
It is an interesting study and topic. Authors have done a tremendous amount of analysis. Looks like data mainly coming from the laboratory. Satisfactory QC tests were performed.
Need following clarifications to make this paper a more scientifically valid paper.
Line 32 – FMN – Need to spell out
Line 78-91 – need to separate method and objectives, Details on selection of sample, what is child check point sites? How representative this sample? Is it a localised or national sample? Need full details about the sample, sampling, representation etc. under method section. Definitions for Socio-Economic Indexes for Areas (SEIFA) and disadvantage Quintile should be provided under the method section. Need to specify how many B vitamins were included under the B vitamers. Type of the population- active, sedentary.
Line 104 - Parents and caregivers provided consent for 105 themselves, what is the role of care giver? Why not consent from parents? Better to clarify. Looks like they have collected samples from caregivers. Better to change wording.
Line 114 – Better to provide reasons for randomisation of samples prior to storing
Line 299 – Sedentary adult need low B vitamers than moderate and heavy active persons. Children age 11-18 years need high level than sedentary adults. Need to explain these in the discussion. Activity level of the adults of this sample. If not known, need to include as limitations. The needs are higher among males than female and active than sedentary as well as children than sedentary adults. Authors have added WHO RDA document in the references list (13), better to discuss the study findings with requirements also. Need to discuss it.
Reviewer 2 Report
The authors have obtained the interesting results that suggest the relationship between family, age, sexe and concentrations of plasma B vitamers in Australian chilndren and adults.
Please consider only these suggestions:
Abstract,
Lines 22-25 The authors indicate: “We aimed to characterise: 1) parent-child dyad concordance, b) generation (children versus adults), c) age (within the adult subgroup of
28-71 y) and d) sex differences in plasma B vitamer concentrations in the
CheckPoint study of Australian children”.
Lines 26-27 “Methods and Results: 1,166 children (11 ± 0.5 y, 51% female) and 1,324
parents (44 ± 5.1 y, 87% female)”
In don’t understand the first paragraph, …28-71, if the age is really 44 ± 5.1y.
Replace FMV by Flavin mononucleotide
Pages 2-5 Materials and Methods,
Line 234. Replace FMV by flavin mononucleotide (FMN)
Pages 5-10 Results
Table 3. Remove Flavin mononucleotide and keep FMN (B2)
Pages 10-12 Discussion,
Line 339 Remove flavin mononucleotide and keep FMN

Reviewer 3 Report
Andraos et al. measured B vitamers in a unique cohort of children and their parents to investigate if familiarity is a relevant determinant of the plasma concentrations of these vitamers. Measurements were performed by a multipley UHPLC/MS-MS method that detects thiamine, riboflavin, FMN, nicotinamide, pantothenic acid and pyridoxid acid. The results show that B vitamer plasma concentrations depend significantly on age and sex, but only little on the inherited genetic background. This is a very interesting study as it helps to interpret individual results and to set appropriate reference ranges. Furthermore, the results provide novel insights into the physiology of the vitamins studied.
Overall, the analyses a unique cohort of relevant size. Data analysis is sound and the paper is well written. Nevertheless, the study has a few weaknesses that should be considered by the authors.
- The cohort should be described in some more detail. Did any of the participants use vitamin supplements? While adults often use such supplements, this is not common in children.
- Did the cohort include individuals with gastrointestinal conditions that may have influenced vitamin absorption?
- Was the us therapeutic drugs considered?
- Why did the authors use a multiplex method with limited sensitivity for individual vitamers rather than targeted methods with optimized conditions for maximum sensitivity and specificity? I think that a substantial number of samples where exact concentrations could not be measured is not ideal.
- The cross-sectional study design with semi-fasting (4h) blood collections is another issue as recent food intake may have influcenced vitamer concentrations in some individuals, but not others. Furthermore, repeat blood collections even a few days apart would have helped to reduce the probabilirty of random effects.
- Why didn’t the authors include other B-vitamins, such as folate and B12?
- Figure 2 should be improved. The lines are barely visible. Furthermore, a dotted vertical line at zero would help to see if confidence intervals include “0”.
Round 2
Reviewer 1 Report
Comments are adopted and ready for publication